# Effectiveness of Machine Learning in Predicting Orthodontic Tooth Extractions: A Multi-Institutional Study

**DOI:** 10.3390/bioengineering11090888

**Published:** 2024-08-31

**Authors:** Lily E. Etemad, J. Parker Heiner, A. A. Amin, Tai-Hsien Wu, Wei-Lun Chao, Shin-Jung Hsieh, Zongyang Sun, Camille Guez, Ching-Chang Ko

**Affiliations:** 1Division of Orthodontics, The Ohio State University, 305 W. 12th Avenue, Columbus, OH 43210, USA; 2College of Dentistry, The Ohio State University, 305 W. 12th Avenue, Columbus, OH 43210, USA; 3Division of Computer Science and Engineering, College of Engineering, The Ohio State University, Columbus, OH 43210, USA; 4Private Practice in Paris, 84200 Carpentras, France

**Keywords:** artificial intelligence, orthodontic tooth extraction, cross-institutional prediction

## Abstract

The study aimed to evaluate the effectiveness of machine learning in predicting whether orthodontic patients would require extraction or non-extraction treatment using data from two university datasets. A total of 1135 patients, with 297 from University 1 and 838 from University 2, were included during consecutive enrollment periods. The study identified 20 inputs including 9 clinical features and 11 cephalometric measurements based on previous research. Random forest (RF) models were used to make predictions for both institutions. The performance of each model was assessed using sensitivity (SEN), specificity (SPE), accuracy (ACC), and feature ranking. The model trained on the combined data from two universities demonstrated the highest performance, achieving 50% sensitivity, 97% specificity, and 85% accuracy. When cross-predicting, where the University 1 (U1) model was applied to the University 2 (U2) data and vice versa, there was a slight decrease in performance metrics (ranging from 0% to 20%). Maxillary and mandibular crowding were identified as the most significant features influencing extraction decisions in both institutions. This study is among the first to utilize datasets from two United States institutions, marking progress toward developing an artificial intelligence model to support orthodontists in clinical practice.

## 1. Introduction

A crucial aspect of orthodontic treatment is determining the appropriate treatment plan, which includes deciding whether to extract or not to extract permanent teeth [1]. Typically, orthodontists base this decision on data from clinical exams, photographs, models, and radiographs, while also relying on their individual training, clinical experience, and treatment philosophies [2,3,4,5,6,7,8,9]. This decision-making process mirrors the principles of artificial intelligence (AI). Historically, there has been limited research on quantifying this decision-making process from a machine-driven perspective to identify the optimal treatment option for each patient. However, with the emergence of AI in medicine and dentistry [10,11], there is now potential to address this long-standing debate more effectively. 

Machine learning, a subset of AI, involves creating algorithms that can learn from data and make decisions based on observed patterns [12]. Several studies have utilized classic machine learning algorithms, such as multi-layer perceptron and random forest (RF), to address the extraction versus non-extraction decision in orthodontics. For instance, Xie et al. developed an artificial neural network with 23 input features, achieving an 80% accuracy rate in predicting whether extraction or non-extraction was the best treatment for 200 malocclusion patients aged 11–15 years [13]. Later, Jung and Kim applied a three-layer neural network to 156 patients treated by a single clinician using 12 cephalometric features and 6 clinical variables as inputs [14]. Their model demonstrated a 93% accuracy rate in the extraction versus non-extraction decision. A similar study employed 24 variables, including demographic, cephalometric, and soft tissue data, and achieved a 94% accuracy rate for the extraction versus non-extraction decision based on 302 patients [15]. In all of these studies [13,14,15], the patient populations were relatively uniform, and the extraction/non-extraction decisions were made by a single orthodontist. 

In contrast, recent studies have explored AI models for predicting tooth extraction in a more diverse patient pool treated by multiple practitioners, aiming for a more robust program with global applicability. Suhali et al. employed a random forest (RF) ensemble classifier for making orthodontic tooth extraction versus non-extraction decisions [16]. Their model achieved a 75% accuracy rate across 287 patients, using 19 diagnostic features, with evaluations conducted independently by five different orthodontists. The authors concluded that the RF model outperformed more complex models, such as neural networks. Additionally, a 2021 study at the University of North Carolina (UNC) advanced previous research by using a larger and more diverse patient pool—838 patients treated by 19 different clinicians, each with their own treatment philosophies—and incorporating varied input data, totaling 117 combined cephalometric and clinical variables [17]. The model’s performance on this sample population, with an accuracy rate of 75–79%, was comparable to the RF study conducted by Suhali et al. [16] 

So far, research on this topic has yielded two main types of findings: models with relatively high accuracy that were developed from uniform datasets, and models with relatively lower accuracy that were derived from more diverse datasets. To advance this field, it is crucial to further develop models that can improve accuracy while still incorporating consecutive patient enrollment of diverse datasets.

The primary aim of this study was to enhance AI models for predicting extraction versus non-extraction decisions in orthodontic patients by evaluating the performance of machine learning on datasets from two universities, University 1 and University 2. A secondary aim was to explore the different treatment philosophies at these two universities by identifying and comparing the rankings of the most important predictive features. Lastly, we plan to test the model using the combined dataset from both institutions.

## 2. Materials and Methods

### 2.1. Dataset

An initial screening was conducted retrospectively on 424 patients who visited the graduate orthodontic clinic at University 1 (Ohio State University) for treatment during consecutive enrollment from 2017 to 2020. Orthodontic treatment for these patients was both initiated and completed within this period. A total of 18 full-time or part-time orthodontic faculty members supervised residents in providing care. All the residents were calibrated through a competency examination to ensure standard diagnosis and treatment planning. To be included in this study, the patients had to have completed orthodontic treatment with a preadjusted edgewise appliance and have complete pre-treatment and post-treatment records in the digital database. The patients who had undergone Phase I treatment, Invisalign treatment, or orthognathic surgery were excluded. Based on these criteria, 297 subjects were recruited, with 247 randomly assigned to the training dataset and the remaining 50 to the testing dataset. The study protocol was reviewed and approved by the university’s Institutional Review Board (IRB# 2020H0513).

An orthodontic resident collected initial records for each patient in the clinic and stored the data in a secure digital database. These records included a clinical exam, intraoral and extraoral photographs, as well as panoramic and lateral cephalometric radiographs. The resident digitized the lateral cephalometric radiographs using University 1’s analysis on the Dolphin Imaging Software 11.95. All the residents were annually calibrated for cephalometric landmark identification through an objective structured clinical examination [18]. A total of 20 combined cephalometric and clinical variables, identified based on previous studies [13,14,15,17], were used in this study. Table 1 provides a list of all the input features along with their definitions. 

The same methods and inclusion/exclusion criteria were applied to the University 2 (University of North Carolina-Chapel Hill) dataset, which also received approval from an Institutional Review Board (IRB# 132183). [17] A total of 838 subjects were recruited, with 695 allocated to the training dataset and the remaining 143 to the testing dataset, which met the training to testing ratio of 83% equivalent to that used in the University 1 model.

### 2.2. Machine Learning Algorithm—Random Forest 

Random forest is a supervised machine learning algorithm that creates and combines multiple decision trees to form a “forest” [19]. It starts at a single point and branches in two directions, with each branch representing a different outcome. The final classification or output is determined by taking the most common prediction from the terminal branches of all the trees [17].

In this study, random forest (RF) was chosen for its excellent performance with tabulated data and its interpretability in various machine learning problems (Figure 1) [20]. The algorithm was implemented using Scikit-learn with the Python programming language, with all the parameters set to default except for the number of trees, which was set to 200 [21].

Three models were developed: Model 1, trained on data from University 1, was used to test data from both institutions; Model 2, trained on data from University 2, was also used to test data from both institutions. Finally, Model 3 was trained using a combined dataset from both universities and tested on a combined testing set. To ensure equal weighting, the training set consisted of 247 randomly selected samples from each university, while the test set included 50 samples from each university. Additionally, feature ranking was calculated by the RF algorithm to determine the importance of each feature in the extraction versus non-extraction decision.

### 2.3. Metrics

The model’s performance was assessed using several metrics: sensitivity (SEN), specificity (SPE), balanced accuracy (BA), accuracy (ACC), positive predictive value (PPV), and negative predictive value (NPV). Table 2 provides definitions for each metric, where TP, TN, FP, and FN stand for true positive, true negative, false positive, and false negative, respectively. In this study, “positive” and “negative” referred to extraction and non-extraction, respectively. The “ground truth” represented the actual orthodontic clinical decision, while “prediction” referred to the AI model’s prediction regarding the extraction or non-extraction decision. The confusion matrix, also known as the error matrix, was provided to allow the validation of the performance of each prediction model. 

## 3. Results

### 3.1. Demographic Distribution

Table 3 shows the demographic distribution for this study. The University 1 dataset included 297 patients, with 93 undergoing extractions and 204 not. The University 2 dataset comprised 838 patients, with 208 having extractions and 630 having non-extractions. Both datasets had a higher proportion of female patients. The average age was comparable across both datasets, and the majority of the subjects were Caucasian, followed by African American, and then Hispanic. 

### 3.2. Performance Comparison among Different Models

To assess the clinical significance of the models, we evaluated accuracy (ACC), which measures each model’s ability to correctly predict extraction versus non-extraction. Given the imbalances in our dataset, we also used balanced accuracy (BA). Model 1 achieved an ACC of 82% and a BA of 74%, while Model 2 had an ACC of 80% and a BA of 64%. When tested with cross-training data, Model 1 predicted on the University 2 data had an ACC of 80% and a BA of 64%. Model 2, when applied to the University 1 data, had an ACC of 75% and a BA of 62%. Sensitivity (SEN), specificity (SPE), positive predictive value (PPV), and negative predictive value (NPV) for these models are detailed in Table 4. Among these two models, Model 1 applied to the University 1 data achieved the highest performance across all metrics: 53% SEN, 94% SPE, 74% BA, 82% ACC, 80% PPV, and 83% NPV.

We also evaluated a model trained on the combined dataset from both institutions (Model 3). When tested on the data from both University 1 and University 2, Model 3 achieved an ACC of 85% and a BA of 74%. Additional metrics for Model 3 are provided in Table 4. The confusion matrices of those five configurations are given in Figure 2.

### 3.3. Feature Rank 

In addition to the previously discussed metrics, the random forest (RF) algorithm can also identify the most important variables for making extraction decisions. Figure 3 and Figure 4 illustrate the variables that the algorithm weighted most heavily when determining extraction versus non-extraction. In machine learning, feature importance ranking (FIR) assesses the contribution of individual input features to the performance of a supervised learning model [22]. 

Figure 3 displays the feature rankings calculated by the random forest (RF) algorithm for the University 1 dataset. The top-ranked variables were (1) maxillary crowding, (2) mandibular crowding, (3) U1-NA (mm) value, (4) SNB (°) value, and (5) U1-NA (°) value. Table 2 provides definitions for all the features used in the study, though only the top five are considered the most important. Figure 4 shows the feature rankings for the University 2 dataset, with the top variables being (1) mandibular crowding, (2) maxillary crowding, (3) L1-NB (mm) value, (4) FMIA (L1-FH) (°) value, and (5) SNB (°) value. In both datasets, the top two variables, maxillary and mandibular crowding, are standard clinical measurements recorded in millimeters, while the next three variables are measurements obtained from the initial cephalometric tracings.

## 4. Discussion

In this study, we assessed the performance of random forest (RF) models for predicting extraction versus non-extraction using data from two university graduate orthodontic clinics. Model 3, which was trained on data from both universities, demonstrated the highest performance across all the metrics, followed by Model 1, which was trained on data from University 1 (see Table 4). This outcome can be attributed to Model 3′s use of combined datasets from both universities, which, as anticipated, enhanced predictive accuracy. Our findings suggest that incorporating multi-center data can effectively improve AI models for predicting orthodontic tooth extraction. 

In all four predictions, we observed low sensitivity (0.29–0.53) and high specificity (0.94–0.96). One reason for the low sensitivity could be the relatively small number of extraction cases in both institutions (31% at University 1 and 25% at University 2), which provided a limited sample size for the model to predict. This result suggests that the models tend to predict non-extraction more frequently than extraction, indicating a conservative approach to predicting extraction cases. Ideally, balancing the training samples by reducing the number of non-extraction cases could improve the model’s performance. However, our dataset was already smaller compared to large-scale examples like autonomous driving tests, and reducing non-extraction cases led to a decrease in accuracy rather than an improvement. Additionally, many orthodontists treat borderline cases—those that could be either extraction or non-extraction—with a non-extraction approach. Thus, the imbalanced training data from our consecutive enrollment reflects real-world orthodontic practices. 

The cross-prediction of data revealed minimal changes in accuracy and a similar sensitivity effect for Model 2 when predicting the University 1 data, while sensitivity decreased by 20% for Model 1 when predicting the University 2 data. These variations in metrics may be attributed to differences in feature distributions in the ground truth. 

We employed the global interpretation method, called feature importance, to examine the interactions between the dependent variable and the independent variables (features) across the entire dataset. This approach evaluates the increase in the model’s prediction error after permuting a feature’s values, thereby disrupting the relationship between that feature and the true outcome. This allowed us to identify the features that influence the model’s decisions. Figure 3 and Figure 4 display the rankings for the University 1 and University 2 datasets, respectively. The top two variables were consistent across both datasets, confirming that both maxillary and mandibular crowding are the most influential factors in making extraction decisions. This consistency helps explain why accuracy remains relatively unaffected. Previous research supports this finding; for example, Li et al. identified “maxillary crowding”, “mandibular crowding”, and “U1-NA” as the key features for the extraction decision. [15]

Interestingly, the next most important variables (3–5) in both datasets pertain to incisor position and inclination. In the University 1 dataset, features 3 and 5 focused on maxillary incisors, specifically U1-NA (mm) and U1-NA (°), respectively. This aligns with Li et al.’s findings, which also highlighted “U1-NA” as a crucial feature for neural network predictions [15]. Conversely, in the University 2 dataset, features 3 and 4 related to mandibular incisors, specifically L1-NB (mm) and L1-FH (°), respectively. Xie et al. similarly emphasized the importance of the lower incisor inclination, specifically “L1-MP”. [13] The discrepancy between the two institutes may be due to their differing treatment philosophies. We observed a slight decrease in cross-predictability when using the University 2 model to predict data from University 1. To improve our understanding in the future, we can employ another interpretation method known as the partial dependence plot. This method focuses on the marginal effect of one or two features on the model’s predicted outcomes. 

These findings suggest that secondary factors contribute to variations in ground truth between institutions, potentially affecting the sensitivity of cross-prediction. This indicates that AI models developed from data from a single institution may not be universally applicable. Future research will focus on combining data from multiple institutions to create an AI model that is more generalizable across different settings.

Combining data from both institutions for the training and testing sets led to improved performance metrics, comparable to or even surpassing those of the models trained on individual institution data. As shown in Table 4, Model 3 demonstrated mild metric increases compared to the previously discussed Models 1 and 2. Specifically, Model 3, trained on the combined dataset from University 1 and University 2, achieved metrics similar to those of Model 1, which was trained solely on University 1 data. This suggests that integrating data from both institutions can sustain the highest performance levels, despite potential differences in treatment philosophies or beliefs between the institutions. 

One limitation of this study was the absence of an outcome assessment to validate our ground truth, making it challenging to confirm the accuracy of the orthodontic clinical decisions. In future research, an internationally accepted index, such as the Peer Assessment Rating (PAR), could be used to evaluate orthodontic treatment outcomes. Additionally, incorporating a multi-expert panel of orthodontists could help standardize the ground truth. Another limitation was the potential influence of hidden or uncollected features. We gathered only 20 inputs (9 clinical features and 11 cephalometric measurements) based on previous studies, but many other factors could impact the clinical decision-making process. For instance, patient or parent beliefs and the soft tissue profile might influence the choice between extraction and non-extraction, yet these were not included in this retrospective study. However, our prior research did examine lip position relative to the E-line using cephalometric data and found that its importance ranking for extraction decisions was not high [17]. Cultural beliefs regarding extraction preferences, which are often not documented in records, may also play a role. Future prospective studies should consider recording and incorporating these additional features into the datasets. 

## 5. Conclusions

In conclusion, the decision between extraction and non-extraction is one of the most challenging clinical choices that orthodontists encounter daily. An effective AI expert system could offer valuable treatment recommendations, helping clinicians verify treatment plans, reduce human error, train orthodontists, and enhance decision-making reliability. [16] AI models have consistently identified crowding as the most critical factor influencing extraction decisions, though other factors may vary depending on the training data source. Combining datasets from multiple institutions can yield performance metrics comparable to those from individual institutions. This study is among the first to apply AI to datasets from two U.S. institutions, marking a significant step toward developing an AI model that could eventually assist orthodontists in clinical practice.

## Figures and Tables

**Figure 1 bioengineering-11-00888-f001:**
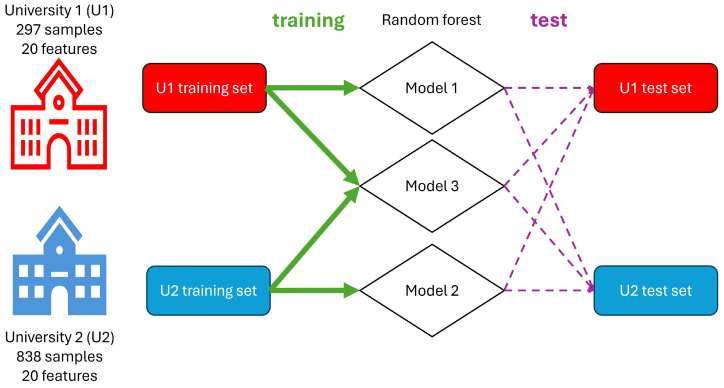
Schematic Workflow. The study involves data collection from two universities. Both datasets (U1 and U2) are divided into training and test sets with the same ratios. Different random forest models (Model 1, Model 2, and Model 3) are trained on the U1, U2, and combined datasets, respectively. The trained models are then evaluated on their respective test sets and cross-applied to the opposite university’s data to assess performance and feature importance. The green arrows represent the training process, the purple dashed arrows represent the testing process, and the diamonds represent the random forest models.

**Figure 2 bioengineering-11-00888-f002:**
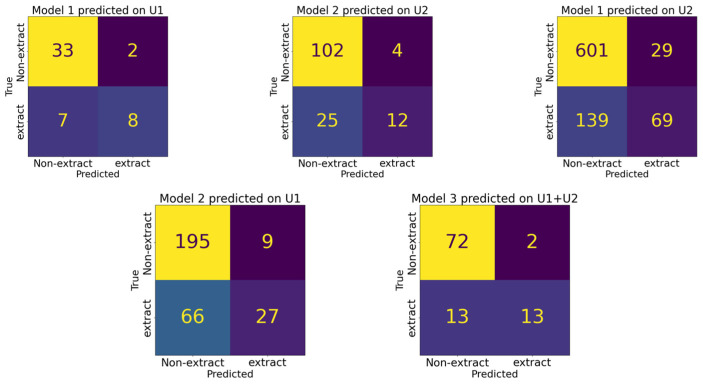
Confusion matrices comparing model predictions across five different configurations. Each matrix represents the model’s performance on a distinct dataset, with true class labels on the *y*-axis and predicted class labels on the *x*-axis.

**Figure 3 bioengineering-11-00888-f003:**
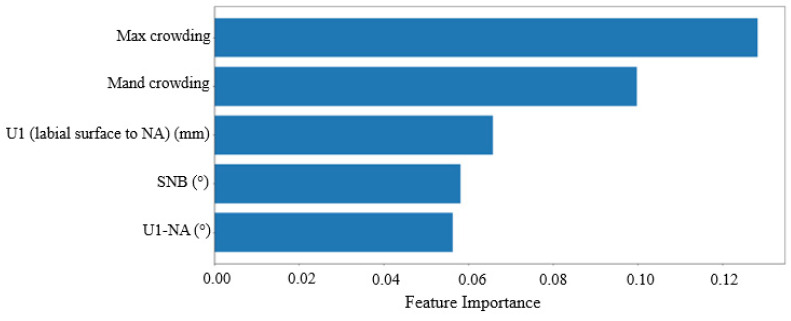
Feature rank calculated by RF for University 1. The *x*-axis represents feature importance and the *y*-axis represents input features (variables). Input features receive a score ranging from 0 to 1, with the sum of all the features equal to 1, and a higher score representing more importance.

**Figure 4 bioengineering-11-00888-f004:**
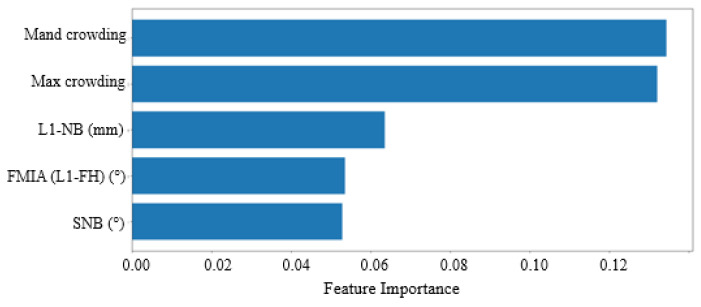
Feature Rank Calculated by RF for University 2. The *x*-axis represents feature importance and the *y*-axis represents input features (variables). Input features receive a score ranging from 0 to 1, with the sum of all features equal to 1, and a higher scorConfirmede representing more importance.

**Table 1 bioengineering-11-00888-t001:** Feature definitions.

Feature	Description of Feature
Gender	Patient biological sex, 0 = female, 1 = male
Age	Patient age at the time of treatment
Overjet (mm) Value	Distance from tip of lower incisor to tip of upper incisor along occlusal plane
Overbite (mm) Value	Distance from the tip of the upper and lower incisor perpendicular to the occlusal plane
Maxillary Crowding (mm)	Amount of maxillary arch crowding
Mandibular Crowding (mm)	Amount of mandibular arch crowding
Molar Classification	Angle classification using two binary variables, class I (0 or 1) and class II (0 or 1); for example, a class III patient would be class I = 0, class II = 0, while a class I patient would be class I = 1, class II = 0
Curve of Spee	The perpendicular distance between the deepest mandibular cusp tip and a flat plane laid on the occlusal surface
SNA (°) Value	Angle created by sella, nasion, and A point
SNB (°) Value	Angle created by sella, nasion, and B point
ANB (°) Value	Angle created by A point, nasion, and B point
U1-NA (°) Value	Angle formed by the long axis of the upper incisor to a line from nasion to A point
U1-NA (mm) Value	Distance between the tip of the upper incisor and a line from nasion to A point
L1-NB (°) Value	Angle formed by the long axis of the lower incisor to a line from nasion to B point
L1-NB (mm) Value	Distance between the tip of the lower incisor and a line from nasion to B point
FMIA (L1-FH) (°) Value	Angle formed by the long axis of the lower central incisor and Frankfort horizontal plane
PFH/AFH (%) Value	The ratio of posterior face height (measured by sella to gonion) to anterior face height (measured by nasion to menton)
FMA (MP-FH) (°) Value	Angle formed by the mandibular plane and Frankfort horizontal plane
Upper Lip to E-Plane (mm) Value	The measurement from the upper lip to the esthetic plane, or a line drawn from the tip of the nose to the tip of the chin

**Table 2 bioengineering-11-00888-t002:** Metric definitions.

Metric	Definition
Sensitivity (TP/TP + FN)	The proportion of clinical extraction cases that were identified by the model correctly
Specificity (TN/TN + FP)	The proportion of clinical non-extraction cases that were identified by the model correctly
PPV (TP/TP + FP)	The proportion of true extraction cases among all the model-predicted extraction cases
NPV (TN/TN + FN)	The proportion of true non-extraction cases among all the model prediction non-extraction cases
ACC (TN + TP/TN + TP + FN + FP)	The proportion of model correctly predicted cases among all the cases
BA (SEN + SPE/2)	The proportion of model correctly predicted cases among all cases adjusted for imbalances in dataset

**Table 3 bioengineering-11-00888-t003:** Demographic distribution of dataset.

	University 1	University 2
Gender		
Male	131 (44.11%)	341 (40.69%)
Female	166 (55.89%)	497 (59.31%)
Age (Mean ± SD)	17.15 ± 8.67	18.37 ± 10.69
Race		
Caucasian	180 (60.61%)	517 (61.70%)
African American	52 (17.51%)	130 (15.51%)
Hispanic	44 (14.81%)	129 (15.39%)
Other	21 (7.07%)	62 (7.40%)
Extraction Type		
Extraction	93 (31.31%)	208 (24.82%)
No-extraction	204 (68.69%)	630 (75.18%)
Total (samples)	297 (247 training, 50 tests)	838 (695 training, 141 tests)

**Table 4 bioengineering-11-00888-t004:** Comparison of performance for University 1 and 2 datasets.

	Accuracy (ACC)	Balanced Accuracy (BA)	Sensitivity (SEN)	Specificity (SPE)	PPV	NPV
Model 1 predicted on University 1 dataset	0.82	0.74	0.53	0.94	0.80	0.83
Model 2 predicted on University 2 dataset	0.80	0.64	0.32	0.96	0.75	0.80
Model 1 predicted on University 2 dataset	0.80	0.64	0.33	0.95	0.70	0.81
Model 2 predicted on University 1 dataset	0.75	0.62	0.29	0.96	0.75	0.75
Model 3 predicted on University 1 and 2 dataset	0.85	0.74	0.50	0.97	0.87	0.85

## Data Availability

The raw data supporting the conclusions of this article will be made available by the authors on request.

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
