# Peer review of "Effectiveness of Machine Learning in Predicting Orthodontic Tooth Extractions: A Multi-Institutional Study"

_bioengineering, 2024, doi:10.3390/bioengineering11090888_

Round 1

Reviewer 1 Report

Comments and Suggestions for Authors

The article explores the application of machine learning in predicting the need for orthodontic tooth extraction. And the study aims to evaluate the performance of machine learning models using data from two different university clinics and identify the most important features for extraction decisions. However, the following points need to be revised and improved:

1. In abstract, the author stated that “Subjects consisted of 297 patients in the Ohio State University…”, but in Materials and methods part, the author mentioned “The training set consisted of a random selection of 247 samples from university 1 and 247 samples from university 2. The test set consisted of 50 samples from university 1 and 50 samples from university 2”. The sample size statement should be accurate and consistent.

2. The introduction part could be more concise and provide a clearer background, research gap and rationale for the study.

3. Please check the format error in all Tables. And Table 3 are unnecessary as it did not provide any related information.

4. In the first paragraph of discussion part, the author stated it “Model 1 predicted on university 1 data set had the highest performance in all metrics. One possible explanation for this result was that different retrieving techniques were used for collection of two data sets…” and “Manual data retrieving may have minimized potential errors encountered with computerized data retrieving”. Actually, the retrieving technique types (manual or computer retrieving) of training data may not be the main reason of accuracy performance.

5. In the second paragraph of discussion part, the author stated that the possible explanation for the low sensitivity was that a small number of extraction cases 220 existed in both institutions. Why didn’t the authors balance the sample ratio of extraction and non-extraction cases in sample collection process?

6. In the third paragraph of discussion part, “These findings may be due to the fact that university 1 data set had a higher proportion of extractions (31% extractions, 69% non-extractions) than the university 2 data set (25% extractions, 75% non-extractions).” Actually the 6% difference of sample ratio selection may not affect the final outcomes.

7. In the fifth paragraph of discussion part, “As shown in Table 5, there is no significant difference in the metrics between model 3 and previously discussed model 1 or 2”. However, there were no any statistical analyses conducted in Table 5.

8. In the last paragraph of discussion part, the author presented several limitations of the study. However, why the author didn’t overcome these limitations? For example, “patient or parent beliefs or soft tissue profile may be important contributing factors for clinical decision on extraction or non-extraction, which were not included in this study”, why the author refuse to include facial soft tissue index as features in the AI models?

9. The language need to be edited.

Comments on the Quality of English Language

The language need to be edited.

Author Response

We have provided detailed point-by-point responses to the comments.

Comment 1: “In abstract, the author stated that “Subjects consisted of 297 patients in the Ohio State University…”, but in Materials and methods part, the author mentioned “The training set consisted of a random selection of 247 samples from university 1 and 247 samples from university 2. The test set consisted of 50 samples from university 1 and 50 samples from university 2”. The sample size statement should be accurate and consistent.”

Response: Thank you for your comments. We acknowledge the deficiency in our methodology regarding the clarity of the sample size in the abstract. While the sample sizes listed in Table 4 are accurate, we have revised the abstract to state: "A total of 1,135 patients, including 297 from University 1 and 838 from University 2, were included during consecutive enrollment periods" (lines 15-16). Additional demographic details for these data sets have been highlighted in lines 91-92 and 107-109.

Comment 2: “The introduction part could be more concise and provide a clearer background, research gap and rationale for the study.”

Response: Thank you for your comments. We have completely rewritten the Introduction section to clarify the background and logic. We have not highlighted the changes, as doing so would be too overwhelming.

Comment 3: “Please check the format error in all Tables. And Table 3 are unnecessary as it did not provide any related information.”

Response: We followed the template for the table format and did not find any errors. However, we agree that Table 3 is unnecessary. Therefore, we have deleted Table 3 and renumbered the remaining tables accordingly.

Comment 4: “In the first paragraph of discussion part, the author stated it “Model 1 predicted on university 1 data set had the highest performance in all metrics. One possible explanation for this result was that different retrieving techniques were used for collection of two data sets…” and “Manual data retrieving may have minimized potential errors encountered with computerized data retrieving”. Actually, the retrieving technique types (manual or computer retrieving) of training data may not be the main reason of accuracy performance.”

Response: We agree that data retrieving method may not have major contributions to the accuracy. We have deleted this discussion and the entire paragraph has been re-written (lines 211-218).

Comment 5: “In the second paragraph of discussion part, the author stated that the possible explanation for the low sensitivity was that a small number of extraction cases 220 existed in both institutions. Why didn’t the authors balance the sample ratio of extraction and non-extraction cases in sample collection process?”

Response: Our retrospective study applied consecutive enrollment with the inclusion and exclusion criteria. The imbalanced data is a natural pattern of orthodontic extraction versus non-extraction samples. Many non-extraction cases are borderline cases, which can go either extraction or non-extraction. Most of orthodontists, patients, and parents incline to non-extraction, which leads the imbalanced data set. If we hand pick cases to force a balanced data set with definitive extraction and definitive extract data, we might get high prediction like Jung and Kim (reference #14) and Li et al. (reference #15). However, these are not generalized models, not applicable to other practitioners. We quickly tested the algorithm by randomly selecting non-extraction patients to match the number of extraction cases; the outcome did not improve the accuracy. We have edited these discussions in the revised paragraph (lines224-231)     

Comment 6: “In the third paragraph of discussion part, “These findings may be due to the fact that university 1 data set had a higher proportion of extractions (31% extractions, 69% non-extractions) than the university 2 data set (25% extractions, 75% non-extractions).” Actually the 6% difference of sample ratio selection may not affect the final outcomes.”

Response: Thank you for the comments. We have removed this stipulation as an explanation. Instead, we rely it to the data features (lines234-235) and moved the discussion of important feature next to this paragraph (lines 236-256).    

Comment 7: “In the fifth paragraph of discussion part, “As shown in Table 5, there is no significant difference in the metrics between model 3 and previously discussed model 1 or 2”. However, there were no any statistical analyses conducted in Table 5.”

Response: Thank you for the comments. Table 5 now is Table 4. We agree that there are no statistical analyses because the mathematical mode predicts one single number without the mean and deviation. As a result, we have removed the word of “significant”. (line 259)    

Comment 8: “In the last paragraph of discussion part, the author presented several limitations of the study. However, why the author didn’t overcome these limitations? For example, “patient or parent beliefs or soft tissue profile may be important contributing factors for clinical decision on extraction or non-extraction, which were not included in this study”, why the author refuse to include facial soft tissue index as features in the AI models?”

Response: This is a retrospective study; some of indices were not labeled in the record. We did test some soft tissue indices in our previous work (reference #17) and they did not make into the top feature list. We have edited these discussions in the revised paragraph (lines 275-278).    

Comment 9: “The language need to be edited.”

Response: Thank you for the comments. The entire manuscript has been re-written.

Reviewer 2 Report

Comments and Suggestions for Authors

Dear Author can you clarify  the comprehensions of  the vantage to use this IA vs Human performance?

Author Response

Comment 1: “Dear Author can you clarify the comprehensions of the vantage to use this AI vs Human performance?”

Response: Thank you for raising this philosophical question. We have completely rewritten the Introduction section to provide a better rationale for our AI studies. In the conclusion, we also provide a statement, adapted from the reference #16, to highlight the advantages of developing an AI system - “An effective AI expert system could offer valuable treatment recommendations, helping clinicians verify treatment plans, reduce human error, train orthodontists, and enhance decision-making reliability.” (lines 283-286)

Round 2

Reviewer 1 Report

Comments and Suggestions for Authors

All tables need to be revised as normative three-line tables.

Other parts have been fine.

Author Response

Comment: All tables need to be revised as normative three-line tables. Other parts have been fine.
- Response: Thanks. We have corrected all tables according to the comments.
